# Speaking Up and Taking Action: Psychological Safety and Joint Problem-Solving Orientation in Safety Improvement

**DOI:** 10.3390/healthcare12080812

**Published:** 2024-04-10

**Authors:** Hassina Bahadurzada, Michaela Kerrissey, Amy C. Edmondson

**Affiliations:** 1Harvard Business School, Harvard University, Soldiers Field Road, Boston, MA 02162, USA; 2Harvard TH Chan School of Public Health, Harvard University, 677 Huntington Ave, Boston, MA 02115, USA

**Keywords:** patient safety, psychological safety, joint problem-solving orientation, clinician turnover, safety climate

## Abstract

Healthcare organizations face stubborn challenges in ensuring patient safety and mitigating clinician turnover. This paper aims to advance theory and research on patient safety by elucidating how the role of psychological safety in patient safety can be enhanced with joint problem-solving orientation (JPS). We hypothesized and tested for an interaction between JPS and psychological safety in relation to safety improvement, leveraging longitudinal survey data from a sample of 14,943 patient-facing healthcare workers. We found a moderation effect, in which psychological safety was positively associated with safety improvement, and the relationship was stronger in the presence of JPS. Psychological safety and JPS also interacted positively in predicting clinicians’ intent to stay with the organization. For theory and research, our findings point to JPS as a measurable factor that may enhance the value of psychological safety for patient safety improvement—perhaps because voiced concerns about patient safety often require joint problem-solving to produce meaningful change. For practice, our conceptual framework, viewing psychological safety and JPS as complementary factors, can help organizations adopt a more granular approach towards assessing the interpersonal aspect of their safety climate. This will enable organizations to obtain a more nuanced understanding of their safety climate and identify areas for improvement accordingly.

## 1. Introduction

Amid considerable progress, improving safety remains a priority in healthcare. Safety events resulting in serious, and sometimes fatal, outcomes are common [1]. Recent estimates in the US indicate that one in four adverse events is preventable [2]. These findings underscore the need to better understand the factors that enable improvements in safety and how to best foster safety in practice.

The frequency and substantial impact of medical errors over many years have inspired work on safety climate in healthcare [3,4]. Safety climate is defined as staff perceptions and attitudes reflecting the priority placed on safety in an organization [5,6]. Safety climate is regarded as key to delivering safe care because perceptions and attitudes about safety shape clinician and staff behaviors in the process of care delivery at a given point in time [5]. Safety climate is considered the surface feature of an underlying safety culture, which comprises the staff’s shared values about the importance of safety, their beliefs about how things operate in their organization, and behavioral norms prioritizing safety [7,8].

Safety climate has effectively been measured in healthcare and has been associated with fewer errors and better patient and clinician outcomes [9,10,11]. This climate is characterized by senior management’s commitment to safety, the use of resources to address safety, the clarity of norms regarding patient safety standards, and interpersonal dynamics among clinicians and staff [5,6]. Healthcare organizations with safety climates are seen to prioritize safety and learning by encouraging frontline staff to adhere to protocols, seek clarification when needed, share insights, report errors and near misses, and engage in learning to enhance safety. Although prior research has explored multiple aspects of safety climates, we lack understanding of how psychological safety and other interpersonal factors lead to material improvements in patient safety.

### 1.1. Background: Psychological Safety and the Surfacing of Safety Concerns

A vital interpersonal aspect of a safety climate is psychological safety—a state of low interpersonal risk that helps people feel able to ask questions, request help, and admit mistakes [12]. Psychological safety has been widely studied in teams across a variety of settings and is associated with beneficial outcomes, such as improved information sharing [13,14], enhanced willingness to voice concerns [15], and enhanced team learning [12,16,17,18]. Psychological safety matters for safety climate because it helps frontline staff voice safety concerns and encourages learning from incidents [19,20]. When people are afraid to speak up about errors, incidents, and near misses, these issues often go undetected, undermining a team’s ability to learn from them and improve safety in the future. In addition to these learning-oriented benefits, frontline staff who perceive their environment as psychologically safe report enhanced wellbeing [21] and reduced emotional exhaustion [22].

Psychological safety is especially needed in the context of healthcare delivery because professional hierarchy and functional diversity, both features of healthcare work, create substantial barriers to speaking up [23,24]. Many good ideas voiced by staff with the intention to improve work processes do not reach fruition because they are rejected by managers and senior leaders without consideration [25]. Other good ideas are embraced initially but become lost in the complex hierarchical structures that characterize most large, multi-layered health systems [26]. Research has found that psychological safety can mitigate the barriers imposed by hierarchy [23], functional diversity [27], and professional boundaries [28]. By reducing the perceived interpersonal risk for raising questions and ideas, psychological safety can help people more easily speak up across professional hierarchies and functional diversity.

### 1.2. Research Gap: From Surfacing Safety Concerns to Addressing Safety Concerns through Joint Problem-Solving

While frontline staff voicing safety concerns and reporting errors matter for improving patient safety, voice alone does not ensure that safety concerns will be addressed [29]. The complexity of healthcare often means that problems leading to safety events and near misses have multiple causes and involve more than one person [30]. Addressing voiced concerns or errors effectively may require diverse functional and technical experts to team up and determine what to do. Recent organizational scholarship points to a need to better understand what occurs after the moment of voice in organizations, especially with team membership fluidity [26]. In fluid and dynamic work settings, collaborative problem-solving can be hard to generate, because individual clinicians and staff work through shifting schedules and rotating patient panels throughout their days [31]. Even well-intentioned individuals willing to raise a concern, admit a mistake, or ask a question may find themselves frustrated and stymied by what happens next if they feel unable to spur changes that might enhance safety.

Recent research on joint problem-solving orientations in fluid teams offers Joint problem-solving orientation (JPS) as one potential factor that may help realize the benefits of psychological safety for patient safety improvement. Joint problem-solving orientation is defined as emphasizing problems as shared and viewing solutions as requiring co-production [32]. Research in hospitals has found that JPS varies significantly across units and that those with higher JPS exhibit greater perceived care quality and safety [33]. The positive relationship between JPS and quality and safety is partially mediated through recognition and appreciation of one another’s skills and knowledge, facilitating the quick surfacing and integration of task-relevant knowledge. This preliminary work developing the concept of JPS in healthcare suggests that it may be relevant to improving safety and complementary to the established role of psychological safety.

### 1.3. New Contribution and Significance of the Study

In this paper we proceed in two steps. First, we expand the existing base of literature on the interpersonal aspect of a safety climate by presenting a conceptual model of psychological safety and joint problem-solving orientation and proposing how, individually and together, they promote safety improvement and worker retention in healthcare. Second, we conduct an exploratory test of these relationships using empirical data from a large healthcare organization in the US.

## 2. Conceptual Model and Hypotheses

We conceptualize psychological safety and joint problem-solving orientation as complementary interpersonal factors that contribute to *patient safety improvement* and frontline staff’s *intent to stay* with an organization. Figure 1 depicts our conceptualization of psychological safety and JPS as complementary by placing each on an axis from low to high, indicating how both can co-occur to a high degree (the upper right-hand quadrant) or a low degree (the lower left-hand quadrant). We suggest, in addition, that individuals may have a mix, represented on the off-diagonal, whereby individuals perceive high psychological safety with little JPS or the converse. These two features may be important to consider together in improving safety because they are each related to improvement behavior and engagement and may interact in ways that further support patient safety.

We proceeded with three main hypotheses based on this model (Figure 2). The first pertains to replicating previous findings associating psychological safety with patient safety [34,35,36], particularly expanding the link to safety *improvement*. To define safety improvement, we drew on process-change literature, which notes that change requires active intervention and choice [37,38]. We view safety improvement as a type of process change in healthcare, one that is defined by active intervention with the aim of improving safety. To make safety improvement, employees must perceive the environment as safe for speaking up. Frontline staff’s input is increasingly valued in quality and safety improvement efforts because they may notice issues that would be otherwise missed [39,40,41]. To offer this input, frontline clinicians and staff must be willing to speak up about what they see [42]. Psychological safety—a state of low interpersonal risk—means frontline staff can speak up without fear of reprisal [12]. Healthcare organizations where frontline staff feel comfortable speaking up demonstrate higher quality decision-making and quicker error detection [43], enhanced quality improvements and safety [44], and increased employee motivation and commitment to the organization [19]. Therefore, we hypothesized that a safe interpersonal climate in which speaking up is expected and encouraged is associated with greater safety improvement and higher intent to stay.

**H1:** 
*Psychological safety is associated with a greater level of safety improvement (H1a) and intent to stay (H1b).*


Our second hypothesis considers the role of joint problem-solving (JPS) in safety improvement. Team membership fluidity in healthcare has been shown to pose challenges to team performance, including safety performance, in part because voicing safety concerns is unlikely to be sufficient to make safety improvements when action is required [26]. Coordinated action to make changes is difficult when team member fluidity is high and differences across functions and expertise lead to the potential for disconnect [45]. JPS describes the perception that challenges are shared, along with a willingness to resolve them together [32,33,46].

Healthcare delivery takes place in a highly specialized environment where timely access to required skills and expertise matters, as does adherence to rules and procedures in pursuit of achieving safe and high-quality care [47,48]. The interdependence of the work, which requires multiple areas of expertise, puts a premium on rapidly establishing awareness of what other expertise domains can contribute. JPS may enhance care safety and quality by helping staff and clinicians understand their interdependence and value the diversity of expertise that others bring [33]. For example, recognizing the value that others offer may lead to more input-seeking behavior, facilitating collaboration in resolving safety concerns and thereby increasing the likelihood of effective safety improvement. We hypothesized that JPS creates a supportive environment, leading to a higher clinician intent to stay and more safety improvement, as the collective crafting of solutions is more likely to reach implementation and enhance safety.

**H2:** 
*Joint problem-solving orientation is associated with a greater level of safety improvement (H2a) and intent to stay (H2b).*


We further posited a positive interaction from the presence of both psychological safety and JPS. When frontline staff speak up and raise concerns and see addressing them as a collective responsibility, this is likely to improve patient care and spur employee engagement in improvement initiatives [32]. Speaking up without support from colleagues for solving problems together may nonetheless occur, stemming from a range of factors, such as professional communication barriers, complex organization structures, power dynamics, or a lack of resources. If these issues are not appropriately addressed, frontline staff may feel ignored or disrespected, prompting them to choose silence in the future [49,50]. Work settings where people feel a degree of JPS but are nonetheless afraid to speak up are also problematic for safety improvement—in these settings, people may expect to collaborate but feel unable to admit errors or report near misses, thus focusing instead on improvement that is less interpersonally risky. We hypothesized that when frontline staff perceive a collective responsibility for addressing safety concerns in their work environment, voiced safety concerns are more likely to be addressed and clinicians will display a higher commitment to the organization.

**H3:** 
*A joint problem-solving orientation positively moderates how psychological safety relates to safety improvement (H3a) and clinician intent to stay (H3b).*


## 3. Methods

### 3.1. Setting and Sample

The data for this research were obtained from a large multi-site health system with their main location in the United States. The health system administers a bi-annual electronic census survey to all employees in English to examine their perception of their work environment. Our data were collected in 2019 (*n* = 42,196, response rate = 87%) and 2021 (*n* = 50,471, response rate = 80%). We only retained respondents for whom we had data on our measures at both time points and excluded all individuals in purely administrative roles, so as to focus on frontline staff (*n* = 14,943). This exclusion was made because frontline staff’s exposure to direct patient care makes their perception of safety especially important for patient safety outcomes [5]. Respondents in our analytical sample identified primarily as physicians, nurses, advanced practice clinicians, or allied health professionals.

### 3.2. Measurement Variables

#### 3.2.1. Independent Variable

*Psychological safety* was measured using four survey items reflecting the extent to which respondents felt safe to speak up when confronted with issues within their organization, as measured in 2019. The items were adapted from the psychological safety scale originally used by Edmondson (1999) [12], with modifications to reflect a particular context of healthcare delivery in which patient safety and patient care are central. The items included (1) “I can report patient safety mistakes without fear of punishment”, (2) “I feel free to raise workplace safety concerns”, (3) “Caregivers will freely speak up if they see something that may negatively affect patient care”, and (4) “Caregivers feel free to question the decisions or actions of those with more authority”. To obtain a comprehensive understanding of the perceived climate, the first two items were measured at the individual level and the latter two items at the organizational level. We calculated Cronbach’s alpha (*a* = 0.79) to assess the internal consistency of the items. We computed the composite measure of psychological safety as a mean of these four variables.

#### 3.2.2. Independent and Moderating Variable

*Joint problem-solving orientation* was adapted from a previously validated measure [32]. Through iterative input with organizational staff, the items measuring joint problem-solving orientation were modified from the original measure in order to reflect the healthcare delivery environment within the organization. It included three items assessing the extent to which employees perceived care delivery to be a collective effort with shared responsibility: (1) “We view addressing problems as a team effort in this department”, (2) “When a problem arises, we routinely involve whomever is needed to address it, regardless of their unit or role”, and (3) “We can rely on people in other departments to address problems with us when needed” (*a* = 0.86). All items used for this measure were captured in 2019.

#### 3.2.3. Dependent Variables

*Safety improvement* examined the extent to which employees believed that safety improvement had been/was being made within their organization, as measured at two time points (2019 and 2021). We operationalized safety improvement as a composite measure of the following two items (calculated as a mean): (1) “In this organization, we are actively doing things to improve patient safety” and (2) “Mistakes have led to positive changes in this organization” (May 2019 *a* = 0.82; May 2021 *a* = 0.84). In line with past work on process change in healthcare [38], we focused on items that emphasized active change toward safety improvement in the organization.

*Intent to stay* was measured by assessing employees’ commitment to remain with the organization. Frontline staff were asked to answer the item “I would stay with this organization if offered a similar position elsewhere” on a five-point Likert scale, ranging from strongly disagree (1) to strongly agree (5).

#### 3.2.4. Control Variables

We obtained and controlled for the following respondent-level characteristics: sex, role, tenure, and race. Sex (0 = male; 1 = female) and race (0 = non-white; 1 = white) were included as binary variables. The respondent’s role and tenure were included as categorical variables with reference categories of physician and tenure of less than one year, respectively.

### 3.3. Statistical Analysis

To create a longitudinal dataset tracking individuals, we merged the frontline staff survey data from 2019 and 2021 at the individual level. We performed descriptive analyses, examining univariate and bivariate statistics for each measure (*n* = 14,943). We calculated Cronbach’s alpha for the composite measures of psychological safety, joint problem-solving orientation, and safety improvement to assess the internal consistency of the items pertaining to each construct. We computed correlations between the independent and dependent variables (Table 1). Joint problem-solving orientation exhibited a correlation of 0.67 with psychological safety, which is consistent with the complementary and reciprocal nature of the constructs.

To examine our hypotheses, we conducted regression analyses, both cross-sectionally in 2019 to examine immediate relationships and longitudinally from 2019 to 2021 to examine the extent to which psychological safety was associated with the moderator and outcomes over time. For the cross-sectional models, all variables were derived from the 2019 survey data; for the longitudinal models, dependent variables were pulled from the 2021 survey data. To test our hypotheses, we ran baseline and interaction regression models at the individual level, clustering the standard errors at the team/department level to adjust for department-level variations (specified as distinct departments in unique locations, e.g., the Emergency Department in Hospital X). We used ordinary least squares (OLS) linear regression models to ease the interpretation of the models; hierarchical linear models yielded broadly consistent results (one moderation finding became slightly more significant; we thus view the OLS results as conservative). Control variables for gender, role, race, and tenure were included based on prior literature associating demographic and status characteristics with psychological safety [23,51,52]. All analyses were conducted using STATA version 18.

## 4. Results

The respondent characteristics of our sample are reported in Table 1. The majority of respondents identified as female (77.55%), and slightly over half reported being a nurse (53.70%). Most had a tenure of between 1 and 10 years (53.06%), with 9.78% reporting a tenure of less than one year and 37.17% a tenure of 11 years or more. Respondents mainly identified as white (81.76%), followed by 7.64% Black or African American, 5.42% Asian, and 3.47% Hispanic or Latino.

Table 2 presents the measure correlations, and Figure 3 presents the unadjusted association between psychological safety and JPS. The scatterplot illustrates how psychological safety and JPS are positively correlated and yet also commonly exhibit observations on the off-diagonal, i.e., with a respondent reporting high on one and low on the other.

Table 3 presents the descriptive statistics for the measures of interest in this study. On average, respondents agreed to experiencing psychological safety (mean = 4.12, SD = 0.71). The mean for JPS was somewhat lower (mean = 3.92, SD = 0.82), on average qualitatively corresponding to not agreeing that they perceived a JPS orientation in their work unit. For the outcome measures, frontline staff perceived more safety improvement in 2019 (mean = 4.31, SD = 0.72) compared to 2021 (mean = 4.21, SD = 0.79), and a similar pattern was present for intent to stay (mean = 4.05, SD=0.97 in 2019; mean = 3.94, SD = 0.85 in 2021).

Table 4 displays the regression results, demonstrating consistent findings across both the cross-sectional and longitudinal models for the baseline and moderation analyses. Psychological safety and joint problem-solving orientation were consistently and statistically significantly associated with safety improvement and intent to stay (*p* < 0.01), in support of Hypotheses 1a and 1b, and 2a and 2b. The presence of higher levels of psychological safety and JPS were both associated with greater safety improvement and intent to stay. Using the cross-sectional models as an example, these relationships can be interpreted as follows: holding other variables constant, a one-point increase in psychological safety was associated with a 0.57-point increase in safety improvement and a 0.39-point increase in intent to stay; a one-point increase in JPS was associated with a 0.13-point increase in safety improvement and a 0.31-point increase in intent to stay.

We also found support for hypotheses 3a and 3b regarding the presence of moderation. The interaction models indicated that psychological safety had a positive significant relationship with safety improvement and that this relationship was stronger in the presence of JPS in both the cross-sectional (β = 0.023, *p* < 0.01) and longitudinal models (β = 0.028, *p* < 0.01). Keeping psychological safety constant, more safety improvement was experienced when respondents took collective responsibility for problems and co-produced solutions. For intent to stay, the moderation analyses demonstrated that the significant main effects of psychological safety and JPS on intent to stay became non-significant in the presence of their interaction. The significant interaction term, indicating a mutually reinforcing relationship between psychological safety and JPS on clinician intent to stay, was observed in both the cross-sectional (β = 0.097, *p* < 0.01) and longitudinal analyses (β = 0.065, *p* < 0.01).

## 5. Discussion and Conclusions

This study sought to understand how psychological safety and joint problem-solving orientation together help alleviate the stubborn challenges that healthcare systems face in seeking to ensure patient safety and reduce clinician turnover. Our analyses show that both psychological safety and JPS relate directly to enhanced safety improvement and clinician commitment to an organization, and we found evidence of moderation, whereby the effect of psychological safety was stronger in the presence of a joint problem-solving orientation. These findings advance our understanding of the interpersonal dynamics for the frontline workers who play a crucial role in influencing safety improvement, both through raising issues and by addressing them effectively together.

In prior research, psychological safety and JPS were shown to predict improvement behavior and engagement, but in this study, we propose a new theory to explain why they may interact in ways that further enhance patient safety. To examine the interaction between psychological safety and JPS, we conceptualized psychological safety and JPS as complementary by placing each on an axis from low to high, indicating how both can co-occur to a high degree (the upper right-hand quadrant) or a low degree (the lower left-hand quadrant). Additionally, individuals may experience a mix, which is represented on the off-diagonal, whereby they experience high psychological safety with little JPS, or the converse. Our findings support that the combined effect of psychological safety and JPS is greater than the sum of their individual effects. When psychological safety and JPS co-occur to a high degree (upper right-hand quadrant), frontline staff are committed to their organization and report greater safety improvement. We posit that the combined presence of JPS and psychological safety is especially effective for safety improvement because it enables issues to be raised *and* addressed together when needed. This builds on past research findings that JPS has both a direct relationship to safety and a relationship that is mediated through enhancing recognition of the value that other expertise areas offer [33]. When healthcare workers feel they can take interpersonal risks to raise issues and then solve them together, they are both better able to concretely address issues, and in the process, they may learn more about one another and how to work to improve and assure day-to-day safety.

There were some notable insights from our findings regarding the off-diagonals. In the quadrant with high psychological safety but low JPS, frontline staff may speak up without receiving input and support from colleagues for addressing the issues that are raised. For safety improvement, our findings suggest that beliefs about speaking up relate positively to patient safety, even independently of this joint support from colleagues (as indicated by the persistent main effect of psychological safety in the moderation model). This is plausible because not all safety concerns require collaborative efforts to be resolved—some are simple issues and/or can be independently addressed. In many healthcare organizations, voiced concerns posing an immediate or critical safety hazard prompt collective action through well-established processes, which are carefully developed and assiduously followed. In these certain instances where the threat is simple and/or quickly addressable, psychological safety is more likely to be sufficient to enhance safety.

In contrast, this persistent relationship between psychological safety and the outcome in the moderation model was not present when considering clinician intent to stay as the outcome. Other research has found that when frontline staff continue to speak up and raise concerns without response, input, and support from colleagues, they can feel ignored or disrespected, decreasing their engagement and exacerbating burnout [32,49]. Our data further emphasize the interdependence between psychological safety and JPS and show a mutually reinforcing relationship between both constructs. Psychological safety and JPS are interrelated in their relationship with clinicians’ commitment to the organization. This was indicated by the statistically significant relationship for the interaction term between JPS and psychological safety and intent to stay.

Studies in other high-pressure environments have shown that improvement efforts tend to be centralized and hierarchical rather than collective and democratic [53,54,55]. This can be an efficient approach to improvement, drawing on the benefits of hierarchical coordination in organizations. However, our measure of psychological safety also points to risks in this approach—specifically, across the items comprising psychological safety, we noted that only half of the respondents reported feeling comfortable to question decisions and actions of those with more authority, a proportion markedly lower than the other items in the psychological safety measure. This indicates that attention to psychological safety in hierarchical safety reporting environments may be especially vital [56].

Our study offers theoretical and practical implications by emphasizing the importance of the interpersonal aspect of safety climates. We introduce JPS as a complementary factor to psychological safety when examining interpersonal dynamics in healthcare. With the introduction of JPS, we urge scholars to adopt a more nuanced approach in understanding how attitudes and cognitions are raised and addressed when needed. This approach can also help managers in healthcare, who can effectively monitor not only frontline staff’s willingness to voice concerns but also their readiness to collaborate in tackling safety challenges together. Tailored solutions and interventions, thus, can be designed and implemented based on whether frontline staff are hesitant to express safety concerns or perceive a lack of collaborative efforts to address reported safety challenges. Therefore, our model provides opportunities to diagnose and improve a team or department.

This study has limitations. First, while we examined how psychological safety and JPS relate to one another and to patient and clinician outcomes, our models indicate associations and not causations. Furthermore, we found some evidence of a dynamic relationship between psychological safety and JPS but did not have access to sufficiently fine-grained longitudinal data to explore it directly. For safety improvement, the interaction coefficient increased over the two-year lag between our cross-sectional and longitudinal models, suggesting that high psychological safety and high JPS mutually reinforce one another over time. Future longitudinal studies can examine the feedback loop between psychological safety and JPS. Second, despite a large analytical sample across different geographic locations, our study was conducted within one large healthcare delivery organization with a shared mission and vision, which may limit the generalizability of our findings across other organizations. Third, we studied the relationship between psychological safety and JPS, but there are likely various other factors, such as hierarchy, human resource management systems, and safety policies, that may interact with psychological safety and JPS. Future research can examine how psychological safety and JPS together interact with other procedural and interpersonal factors to affect patient and clinician outcomes.

Our study emphasizes the importance of the interpersonal aspects of a safety climate in enhancing patient and clinician outcomes in healthcare. Interpersonal dynamics are important in all sectors, but in healthcare—a highly specialized sector—where effective collaboration and timely access to required skills and expertise save lives, interpersonal dynamics are vital to the quality and safety of care delivery.

## Figures and Tables

**Figure 1 healthcare-12-00812-f001:**
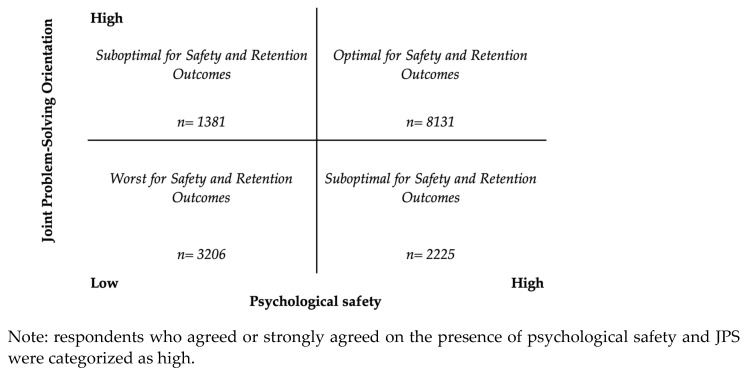
Psychological safety and joint problem-solving orientation.

**Figure 2 healthcare-12-00812-f002:**
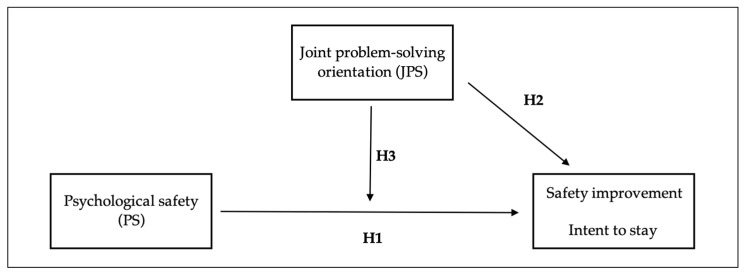
Hypothesized research model.

**Figure 3 healthcare-12-00812-f003:**
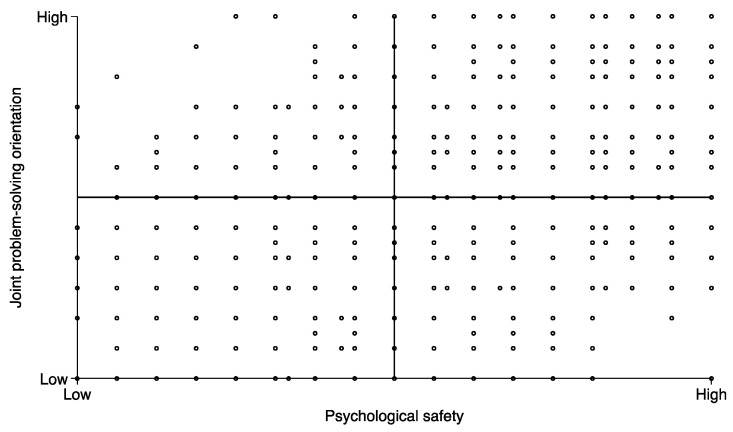
The relationship between psychological safety and JPS.

**Table 1 healthcare-12-00812-t001:** Sample characteristics (*n* = 14,943).

Characteristics	*n* (%)
Female	11,589 (77.55%)
Role	
Physician	2214 (14.82%)
Nurse	8024 (53.70%)
Advanced practice clinician	1019 (6.82%)
Allied health professional	3686 (24.67%)
Race	
White (not of Hispanic origin)	12,217 (81.76%)
Black or African American	1142 (7.64%)
Asian	810 (5.42%)
Hispanic or Latino	519 (3.47%)
Other	255 (1.71%)
Tenure	
Less than 1 year	1461 (9.78%)
Tenure 1–10 years	7929 (53.06%)
Tenure 11–20 years	3490 (23.36%)
Tenure >20 years	2063 (13.81%)

**Table 2 healthcare-12-00812-t002:** Correlations between independent and dependent measures.

	Psychological Safety (2019)	Joint Problem-Solving (2019)	Safety Improvement (2019)	Safety Improvement (2021)	Intent to Stay (2019)	Intent to Stay (2021)
Psychological safety (2019)	1	0.67 *	0.66 *	0.37 *	0.50 *	0.31 *
Joint problem-solving (2019)		1	0.52 *	0.33 *	0.49 *	0.30 *
Safety improvement (2019)			1	0.43 *	0.46 *	0.31 *
Safety improvement (2021)				1	0.29 *	0.49 *
Intent to stay (2019)					1	0.44 *
Intent to stay (2021)						1

Note: * corresponds to *p*-value < 0.01.

**Table 3 healthcare-12-00812-t003:** Measure descriptions: *n*, mean, standard deviation (SD), and response distribution.

Measures	*n*	Mean	SD	Response Distribution (%)				
				1	2	3	4	5
Psychological safety (2019)	14,936	4.12	0.71					
Report patient safety mistakes without fear of punishment	14,671	4.37	0.85	1.45	2.91	6.94	34.94	53.77
Feel free to raise workplace safety concerns	14,886	4.32	0.82	1.07	2.81	7.78	40.20	48.15
Caregivers speak up if something negatively affects patient care	14,759	4.29	0.85	1.16	3.69	8.02	39.64	47.49
Caregivers feel free to question those with more authority	14,752	3.53	1.09	4.77	13.72	23.67	39.19	18.65
Joint problem-solving orientation (2019)	14,606	3.92	0.82					
View addressing problems as a team effort	14,547	3.94	0.96	1.97	7.34	15.40	45.25	30.05
Involve whomever needed to address the problem	14,416	3.99	0.91	1.65	5.69	14.73	47.96	29.97
Rely on people in other department to address problems with us	14,157	3.83	0.90	1.53	6.56	21.83	47.51	22.58
Safety improvement (2019)	14,855	4.31	0.72					
We are actively doing things to improve patient safety	14,818	4.38	0.77	0.88	1.88	7.22	38.39	51.63
Mistakes have led to positive changes	14,586	4.24	0.79	0.73	1.89	12.42	42.80	42.17
Safety improvement (2021)	14,865	4.21	0.79					
We are actively doing things to improve patient safety	14,834	4.27	0.86	1.51	2.96	9.30	39.31	46.93
Mistakes have led to positive changes	14,620	4.15	0.85	1.07	2.81	14.63	42.59	38.90
Intent to stay (2019)	14,790	4.05	0.90	1.16	3.56	20.22	39.67	35.38
Intent to stay (2021)	14,763	3.94	0.97	2.08	5.03	22.62	36.97	33.30

**Table 4 healthcare-12-00812-t004:** Models relating psychological safety (PS) and joint problem-solving (JPS) to safety improvement and intent to stay.

	Safety Improvement				Intent to Stay			
	Cross-Sectional		Longitudinal		Cross-Sectional		Longitudinal	
	Main Effect	Interaction	Main Effect	Interaction	Main Effect	Interaction	Main Effect	Interaction
Psychological safety (PS)	0.565 ***	0.485 ***	0.286 ***	0.186 ***	0.392 ***	0.049	0.276 ***	0.046
Joint problem-solving (JPS)	0.131 ***	0.040	0.162 ***	0.048	0.305 ***	−0.086 *	0.199 ***	−0.062
PS # JPS		0.023 **		0.028 ***		0.097 ***		0.065 ***
Female	0.018	0.020 *	0.023	0.025	0.028 *	0.035 **	0.026	0.030
Role								
Advanced practice provider	−0.040	−0.039 *	−0.148 ***	−0.146 ***	0.086 ***	0.092 ***	−0.054	−0.050
Nurse	−0.082 ***	−0.080 ***	−0.242 ***	−0.240 ***	0.143 ***	0.150 ***	−0.049 *	−0.044
Allied health professional	−0.083 ***	−0.082***	−0.185 ***	−0.184 ***	0.171 ***	0.177 ***	−0.029	−0.025
Tenure								
1–10 years	−0.007	−0.008	0.127 ***	0.126 ***	−0.075 ***	−0.079 ***	0.149 ***	0.147 ***
11–20 years	0.025	0.023	0.223 ***	0.221 ***	−0.030	−0.038	0.306 ***	0.301 ***
>20 years	0.073 ***	0.071 ***	0.278 ***	0.275 ***	0.060 **	0.052 **	0.439 ***	0.434 ***
White	0.011	0.012	−0.029	−0.028	−0.039 **	−0.035 **	0.035 *	0.038 *
Intercept	1.498 ***	1.807 ***	2.432 ***	2.820 ***	1.156 ***	2.485 ***	1.800 ***	2.690 ***
Number of observations	14,549	14,549	14,545	14,545	14,474	14,474	14,438	14,438
R-squared	0.45	0.45	0.17	0.17	0.30	0.31	0.13	0.13

Note: * corresponds to *p*-value < 0.10, ** corresponds to *p*-value < 0.05, and *** corresponds to *p*-value < 0.01.

## Data Availability

The data for this study were made accessible to the authors by the organization through a data use agreement. The data are not publicly available. Requests for further information about the data can be directed to mkerrissey@hsph.harvard.edu.

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
