# Peer review of "Speaking Up and Taking Action: Psychological Safety and Joint Problem-Solving Orientation in Safety Improvement"

_healthcare, 2024, doi:10.3390/healthcare12080812_

Round 1

Reviewer 1 Report

Comments and Suggestions for Authors

First, I would like to convey my gratitude to the authors for their wonderful piece of work and their patience. Upon careful review of the paper, the subsequent remarks are a few suggestions for additional refinement.

1) The introduction section was very informative about the study variables. In any empirical investigation, we do use any sub-section under the introduction section. The introduction section covers the background of the study, research gaps and novelty, study objectives, and the significance/importance of the study. Accordingly, it is requested to follow the stated structure if possible, otherwise, just add the research gap, newness/uniqueness of the study, and significance in sub-section 1.3 (overview). Presently, all these issues have not been addressed. 

2) A very well-written literature review. 

3) Under the subheading "Settings and Sample" in the method section, please describe how you went about selecting each respondent. What sampling technique have you used to select them? What was the total population? What was the representative sample size? Also please explain the existing sample size (14,943).

4) Please provide further elaboration on the process of data collection.

5) Sub-section 3.2 should be the measurement of variables. Under this sub-section, please discuss the details of all three variables i.e., independent, moderating, and dependent. Also please mention "Joint problem-solving orientation" as a moderating variable.

6) It is also requested to cross-check the descriptions of Table 2 and Table 3 (refer to lines 267 and 279).

7) Please add research implications based on study findings, both theoretical and practical/managerial implications. Presently, the implications/contributions are not discussed separately and may be precisely covered in the discussion section.

Thank you and best wishes.

Author Response

Dear Reviewer,

Reviewer 1

First, I would like to convey my gratitude to the authors for their wonderful piece of work and their patience. Upon careful review of the paper, the subsequent remarks are a few suggestions for additional refinement.

  • 1) The introduction section was very informative about the study variables. In any empirical investigation, we do use any sub-section under the introduction section. The introduction section covers the background of the study, research gaps and novelty, study objectives, and the significance/importance of the study. Accordingly, it is requested to follow the stated structure if possible, otherwise, just add theresearch gap, newness/uniqueness of the study, and significance in sub-section 1.3 (overview). Presently, all these issues have not been addressed. 

Response: To follow the suggested sections background, research gap and novelty, study objectives, and significance, we have made the following edits:

  • We added text on page 2 (see track changes) to indicate the research gap and significance. In this paper we argue that the interpersonal aspect of a safety climate requires special attention in healthcare. We adopt a more granular approach to understand how safety concerns are voiced and addressed (research gap). Considering the high interdependence and highly specialized nature of care delivery, we argue that a more granular understanding of the interpersonal dynamics will help us identify how we can enhance safety by making sure that perceived safety concerns are voiced and addressed effectively (significance).
  • Under sub-section 1.3, we added the following sentence to indicate the new/uniqueness of the study: We expand the existing base of literature examining the interpersonal aspect of a safety climate by presenting a conceptual model of psychological safety and joint problem-solving orientation and proposing how they individually and together promote safety improvement and worker retention in health care.

With regard to subheadings used in the introduction, this paper is mainly presenting a new framework that is being tested using data from a large healthcare system. We introduce psychological safety and joint problem-solving orientation (JPS) as complementary factors in enhancing safety. Considering that the introduction is relatively long, the subheadings are used to provide structure that will help readers understand how psychological safety and JPS affect patient and provider outcomes individually and together.

  • 2) A very well-written literature review. 

Response: We appreciate the kinds word, thank you very much.

  • 3) Under the subheading "Settings and Sample" in the method section, please describe how you went about selecting each respondent. What sampling technique have you used to select them? What was the total population? What was the representative sample size? Also please explain the existing sample size (14,943).

Response: Our data comes from a bi-annually administered survey to all organizational employees. In 2019, 42,196 employees completed the survey which translated to an 87% response rate. In 2021, 50,471 employees completed the electronic survey (response rate = 80%). As this survey was a census of the whole organization, there was no sampling (i.e., the survey was sent to all employees). We have added text in section 3.1 to clarify that the survey was intended to take a census.

We merged both datasets at the individual level, only including employees for whom we had data on all measures in both years. Additionally, we excluded all employees in purely administrative roles to test our hypothesis with patient-facing staff. This yielded an analytic sample of N = 14,943 that is a demographically representative sample of the full sample in both years.

We included the sample the total sample in each year under subheading Settings and Sample.

  • 4) Please provide further elaboration on the process of data collection.

Response: The survey tool is administered by the healthcare system itself. The dataset represents a bi-annual administered survey by the organization to examine employees’ perception of their work environment. The aim is to get as many responses as possible, drawing from a census of all employees. This is reflected in the high response rate. 

  • 5) Sub-section 3.2 should be the measurement of variables. Under this sub-section, please discuss the details of all three variables i.e., independent, moderating, and dependent. Also please mention "Joint problem-solving orientation" as a moderating variable.

Response: Thank you for bringing this to our attention, we have added the sub-section Measurement variables and indicated JPS as both independent as well as moderator variable.

  • 6) It is also requested tocross-check the descriptions of Table 2 and Table 3 (refer to lines 267 and 279).

Response: Thank you for bringing this to our attention. We have corrected this.

  • 7) Please add research implications based on study findings, both theoretical and practical/managerial implications. Presently, the implications/contributions are not discussed separately and may be precisely covered in the discussion section.

Response: Theoretical: in this study, we emphasize the importance of the interpersonal aspect of safety climates and introduce JPS as a complementary factor to psychological safety in examining interpersonal dynamics in the healthcare setting. With the introduction of JPS, we urge scholars to adopt a more nuanced approach in understanding how attitudes and cognitions affect the extent to which safety concerns are raised and addressed together when needed.  

Practical: to address the ongoing challenges in patient safety, managers will benefit from a comprehensive understanding of interpersonal dynamics in healthcare teams. This will enable managers to effectively monitor not only frontline staff’s willingness to voice concerns but also their readiness to collaborate in tackling safety challenges together. Depending on whether frontline staff are reluctant to voice safety concerns or rather voice concerns but perceive a lack of collaborative efforts in solving the reported safety challenges, the solutions and interventions required can be tailored to the team’s needs.

Reviewer 2 Report

Comments and Suggestions for Authors

·        In the Abstract the goal of the paper must be emphasized.

·        Fig. 1 on vertical axis JPS also has to be marked  as low and high \

·        -At row 187 – there is another type of citation ((Singer, 2009);

·        In limitations of the research has to be mentioned that HRM system has impact on JPS , employees‘ safety and team work, but is not discussed in this paper;

·        There is a self citations of Michaela Kerrissey  and Amy C. Edmondson

Author Response

Dear Reviewer,

Reviewer 2

  • In the Abstractthe goal of the paper must be emphasized. 

Response: We have added the following text to explicitly state the goal of the paper: This paper aims to advance theory and research on patient safety by elucidating how the role of psychological safety in patient safety can be enhanced with joint problem-solving orientation (JPS).   

  • 1 onvertical axis JPS also has to be marked as low and high 

Response: Thank you for bringing this to our attention, we have added labels to the JPS axis.

  • At row 187 – there is another type of citation ((Singer, 2009);

Response: Thank you for bringing this to our attention, we corrected the citation conform the MDPI citation style.

  • In limitations of the research has to be mentioned that HRM system has impact on JPS, employees‘ safety and team work, but is not discussed in this paper;

Response: Thank you for your suggestion, we have included human resource management as a factor that may affect psychological safety and JPS in the limitations section. In this study, we bring forward JPS as a complementary factor to psychological safety and we agree that future research needs to look into how JPS relates to various aspect of human management and social structures within healthcare organizations.

  • There is a self citations of Michaela Kerrisseyand Amy C. Edmondson

Response: We appreciate and acknowledge the importance of adhering to guidelines regarding self-citation. As such, we have made diligent efforts to minimize self-citation, reducing the count to ten. The challenge in reducing self-citation in this paper stems from the introduction of a novel conceptual framework for psychological safety and joint problem-solving orientation. These concepts were first introduced into healthcare by Prof. Amy Edmondson and Prof. Michaela Kerrissey. This explains the number of self-citations and the challenge in replacing them without compromising our ability to present a comprehensive conceptual framework. We have corresponded about this matter with the editor as well.

Reviewer 3 Report

Comments and Suggestions for Authors

The article is promising in terms of the originality of the psychological safety and joint problem-solving orientation, as evidenced by the hypothesized properly tested research model. However, to realize its full potential, the article should be edited; provide further evidence and arguments to support its relevance, methods, findings, discussion, and conclusion.

1. The abstract is too long. The purpose of the study is a unclear testable hypothesis. The methods lack clearly design, sampling, and statistical testing (see line 10-24).

2. Introduction lack clearly indemnified the problem statement, existing in testing factor, and objectives. The introduction is too short and lack evidentiary support. What is a research gap of testing variables? How do factors deduct from theoretical framework? What is the main purpose of the study? These issues are uncleared (see line 29-49).

3. Review is uncleared how do authors deductively definitely factor/variables from the theory. The review is too general literature, which lacks deeply clarified psychological safety affected on problem-solving orientation and safety improvement (see line 50-98).

4. Methods unclearly approached setting and sample. Where are 14,943 samples coming from? What are the sources of samples? How did authors select the samples? Are samples represented all groups? Should be clearly clarified why?

5. The results lack ordered evidence. The result is missing a presentation. The authors never go back to test the hypothesis, why?

6. Discussion should be separated between discussion and conclusion. It is uncleared if the done not discuss with the main results, especially hypothesis testing results, why? Should be strictly followed the main discussion with hypothesis results. And then provided the practical, theoretical, and policy implications are required.

Comments on the Quality of English Language

Minor editing is required. Some sentences are invalid communication and meaning.

Author Response

Dear Reviewer,

Reviewer 3

The article is promising in terms of the originality of the psychological safety and joint problem-solving orientation, as evidenced by the hypothesized properly tested research model. However, to realize its full potential, the article should be edited; provide further evidence and arguments to support its relevance, methods, findings, discussion, and conclusion.

  • 1) The abstract is too long. The purpose of the study is a unclear testable hypothesis. The methods lack clearly design, sampling, and statistical testing (see line 10-24).

Response: Thank you for raising these points. We have made the following changes to further clarify the study:

  • Purpose of the study: we have added sentences in the abstract and the introduction clearly stating the purpose of the study. Furthermore, we have changed the subheadings in the introduction such that the background, research gap, study objectives, and contribution of the paper are made more explicit.
  • Sampling: our data comes from a bi-annually administered survey to all organizational employees. In 2019, 42,196 employees completed the survey which translated to an 87% response rate. In 2021, 50,471 employees completed the electronic survey (response rate = 80%). As this survey was a census of the whole organization, there was no sampling (i.e., the survey was sent to all employees).

We merged both datasets at the individual level, only including employees for whom we had data on all measures in both years. Additionally, we excluded all employees in purely administrative roles to test our hypothesis with patient-facing staff. This yielded an analytic sample of N = 14,943 that is a demographically representative sample of the full sample in both years.

We have added text in section 3.1 to clarify that the survey was intended to take a census and included the total sample in each year under subheading Settings and Sample.

  • Design and statistical testing: our study is an observational study in which we ran baseline and interaction regression models at the individual level, clustering the standard errors at the team/department level to adjust for department level variation We used ordinary least squares (OLS) linear regression models to ease the interpretation of the models. If this clarification does not adequately address the concern, please do not hesitate to let us know. We are happy to address any further comments or concerns regarding the design and statistical testing.

  • 2) Introduction lack clearly indemnified the problem statement, existing in testing factor, and objectives. The introduction is too short and lack evidentiary support. What is a research gap of testing variables? How do factors deduct from theoretical framework? What is the main purpose of the study? These issues are uncleared (see line 29-49).

Response: We have made the following changes to address concern regarding the research gap, deducting factors from the theoretical framework, and purpose of the study:

  • Research gap: in addition to changing the subheadings in the introduction to elucidate the research gap, we added the following sentence; While existing literature has explored multiple aspects of safety climates, a granular understanding of how interpersonal aspects such as psychological safety can lead to material improvements in patient safety over time has been lacking to present the research gap more explicitly.
  • Factors deducted from theoretical framework: the text used to describe the four quadrants in the two-by-two table are not meant as introducing new factors and are not meant as a comprehensive theoretical model; rather, they are meant to point to two interrelated factors that we hypothesize may play an important role together in helping to improve patient safety. The table and the text within the quadrant are intended to provide an intuitive understanding of how high/low psychological safety/joint problem-solving orientation relate to patient and provider outcomes.
  • Purpose of the study: please see clarification under comment 1.

  • 3) Review is uncleared how do authors deductively definitely factor/variables from the theory. The review is too general literature, which lacks deeply clarified psychological safety affected on problem-solving orientation and safety improvement (see line 50-98).

Response: Psychological safety and JPS are considered complementary factors affecting the interpersonal aspects of a safety climate. In the introduction we describe the relation between psychological safety and JPS and safety improvement. Their relationships are discussed after line 98.

  • 4) Methods unclearly approached setting and sample. Where are 14,943 samples coming from? What are the sources of samples? How did authors select the samples? Are samples represented all groups? Should be clearly clarified why?

Response: We have clarified the methods in the document. The primary data collection is conducted by the healthcare system itself. The dataset represents a bi-annual administered survey by the organization to examine employees’ perception of their work environment. For more details, please see the response to comment 1.

  • 5) The results lack ordered evidence. The result is missing a presentation. The authors never go back to test the hypothesis, why?

Response: In the introduction, we introduce three hypotheses which are reported under results and further discussed in the discussion:

  • H1: Psychological safety is associated with a greater level of safety improvement (H1a) and intent to stay (H1b).
  • H2: Joint problem-solving orientation is associated with greater level of safety improvement (H2a) and intent to stay (H2b).
  • H3: A joint problem-solving orientation positively moderates how psychological safety relates to safety improvement (H3a) and clinician intent to stay (H3b).

In the study, we tested these three hypotheses and the results are reported, please see line 355-369. We have also copy-pasted the results below to ease your review.

“Psychological safety and joint problem-solving orientation were consistently and statistically significantly associated with safety improvement and intent to stay (p<0.01), in support of Hypotheses 1a and 1b, and 2a and 2b. The presence of higher levels of psychological safety and JPS are both associated with greater safety improvement and intent to stay. Using the cross-sectional models as an example, these relationships can be interpreted as follows: holding other variables constant, a one-point increase in psychological safety is associated with a 0.57-point increase in safety improvement and a 0.39-point increase in intent to stay; a one-point increase in JPS is associated with a 0.13-point increase in safety improvement and a 0.31-point increase in intent to stay.

We also found support for hypotheses 3a and 3b regarding the presence of moderation. The interaction models indicate that psychological safety has a positive significant relationship with safety improvement, and that this relationship is stronger in the presence of JPS in both the cross-sectional ( = 0.023, p<0.01) and longitudinal models ( = 0.028, p<0.01).”

  • 6) Discussion should be separated between discussion and conclusion. It is uncleared if the done not discuss with the main results, especially hypothesis testing results, why? Should be strictly followed the main discussion with hypothesis results. And then provided the practical, theoretical, and policy implications are required.

Response: Thank you for bringing this to our attention. We have added the following paragraph to the discussion stating the theoretical and practical implication of our study: Our study offers theoretical and practical implications by emphasizing the importance of the interpersonal aspect of safety climates. We introduce JPS as a complementary factor to psychological safety when examining interpersonal dynamics in healthcare. With the introduction of JPS, we urge scholars to adopt a more nuanced approach in understanding how attitudes and cognitions are raised and addressed when needed. This approach can also help managers in healthcare, who can effectively monitor not only frontline staff’s willingness to voice concerns but also their readiness to collaborate in tackling safety challenges together. Tailored solutions and interventions thus can be designed and implemented based on whether frontline staff are hesitant to express safety concerns or perceive a lack of collaborative efforts to address reported safety challenges. Our model thus provides opportunities to diagnose and improve a team or department.

For conclusions, please see the first paragraph of the discussion where we represent the conclusion to our three hypotheses (284-287). 

Round 2

Reviewer 3 Report

Comments and Suggestions for Authors

As  current revised version is well-suited publication in the healthcare. Good luck!

Comments on the Quality of English Language

Moderate editing required.